# Macrophage Lysosomal Alkalinization Drives Invasive Aspergillosis in a Mouse Cystic Fibrosis Model of Airway Transplantation

**DOI:** 10.3390/jof8070751

**Published:** 2022-07-20

**Authors:** Efthymia Iliana Matthaiou, Wayland Chiu, Carol Conrad, Joe Hsu

**Affiliations:** 1Department of Medicine, Division of Pulmonary, Allergy and Critical Care Medicine, Stanford University School of Medicine, Stanford, CA 94304, USA; ematth@stanford.edu (E.I.M.); chiu@icahn.mssm.edu (W.C.); 2Icahn School of Medicine at Mount Sinai, New York, NY 10029, USA; 3Department of Pediatrics, Pulmonary Medicine, Stanford University School of Medicine, Stanford, CA 94304, USA; cconrad@stanford.edu

**Keywords:** cystic fibrosis, airway transplantation, invasive aspergillosis, macrophage alkalinization

## Abstract

Cystic fibrosis (CF) lung transplant recipients (LTRs) exhibit a disproportionately high rate of life-threatening invasive aspergillosis (IA). Loss of the cystic fibrosis transmembrane conductance regulator (CFTR^-/-^) in macrophages (mφs) has been associated with lyosomal alkalinization. We hypothesize that this alkalinization would persist in the iron-laden post-transplant microenvironment increasing the risk of IA. To investigate our hypothesis, we developed a murine CF orthotopic tracheal transplant (OTT) model. Iron levels were detected by immunofluorescence staining and colorimetric assays. *Aspergillus fumigatus* (*Af*) invasion was evaluated by Grocott methenamine silver staining. Phagocytosis and killing of *Af* conidia were examined by flow cytometry and confocal microscopy. pH and lysosomal acidification were measured by LysoSensor^TM^ and Lysotracker^TM^, respectively. *Af* was more invasive in the CF airway transplant recipient compared to the WT recipient (*p* < 0.05). CFTR^-/-^ mφs were alkaline at baseline, a characteristic that was increased with iron-overload. These CFTR^-/-^ mφs were unable to phagocytose and kill *Af* conidia (*p* < 0.001). Poly(lactic-co-glycolic acid) (PLGA) nanoparticles acidified lysosomes, restoring the CFTR^-/-^ mφs’ ability to clear conidia. Our results suggest that CFTR^-/-^ mφs’ alkalinization interacts with the iron-loaded transplant microenvironment, decreasing the CF-mφs’ ability to kill *Af* conidia, which may explain the increased risk of IA. Therapeutic pH modulation after transplantation could decrease the risk of IA.

## 1. Introduction

*Aspergillus fumigatus* (*Af*) is a ubiquitous mold that grows in decaying organic matter, releasing airborne spores that results in an estimated 20 million cases of respiratory and sinus diseases worldwide [1]. *Aspergillus*-related respiratory diseases include severe asthma, tracheobronchitis, chronic necrotizing *Aspergillus* pneumonia and invasive pulmonary aspergillosis (IPA) [2,3,4,5,6,7,8,9,10]. These diseases occur in immune-suppressed persons and in those with underlying pulmonary comorbidities such as cystic fibrosis (CF), asthma and chronic obstructive pulmonary disease (COPD) [10,11,12,13]. These infections are particularly common among lung transplant recipients (LTRs), with one-in-three persons suffering from an *Af*-related pulmonary disease [14,15]. In addition to IPA, CF-lung transplant recipients (CF-LTRs) are at a highest risk for *Af* infections of the tracheal anastomosis, resulting in serious complications including fatal hemorrhage, airway stenosis and bronchomalacia [16,17]. In invasive disease, *Af* spores germinate into filamentous hyphae and invade and destroy tissues and organs, with mortality rates as high as 60% [18]. There are only a limited number of antifungal treatments and there is growing *Af* resistance to these medications [19,20,21]. The development of successful therapeutics to fight these infections requires a more comprehensive understanding of the interplay of immune mechanisms that control *Af* in vivo.

In CF-LTRs, the incidence of invasive *Af* exceeds that seen for transplants undertaken for non-CF-related pulmonary diseases [22,23]. Putative factors such as fungal colonization prior to transplantation do not completely explain this increased risk [24]. One factor that has been suggested to increase the risk of infections in persons with CF is innate immune defects caused by the loss of the cystic fibrosis transmembrane conductance regulator (CFTR^−/−^) [25,26]. CFTR is a cellular receptor that mediates Cl^-^ entry and functions as counter-ion conductase to balance the H^+^ influx to maintain an acidified lysosomal environment [27]. CFTR is highly expressed by several types of lung epithelial cells, as well as neutrophils, monocytes and macrophages, underlying the potential importance of CFTR deficiency in the innate immune response [28]. Mφs are central to the host defense of *Af* through phagocytosis and killing of conidia as well as by directing the host pro-inflammatory response that recruits neutrophils to the site of infection [29]. Mφs can recognize *Af* conidia by pattern recognition receptors such as Toll-like receptor (TLR)-2, TLR-4 and dectin-1 [30,31] and engulf them. The conidia are then transferred into the phagolysosome (mature phagosome that has been acidified by lysosomes), where the conidia are broken down by enzymes that are active only in an acidic microenvironment [32]. Several studies have shown that CFTR-deficiency alters mφs’ signaling pathways, and directly controls CF-mφs’ lysosomal pH and microbicidal function [33,34,35,36,37,38,39,40,41]. Other studies have failed to replicate these findings possibly due to differences in experimental procedures, including methods used to measure pH and the use of live organisms versus opsonized particles [27,42,43]. In this study, our goal was to clarify the relationship between the loss of CFTR and the mφ’s ability to clear *Af* conidia in the setting of transplantation, using a murine orthotopic tracheal transplant (OTT) model [17,44].

We and others have previously shown that the human transplanted lung is characterized by iron-overloaded mφs [44,45,46]. In addition to their anti-microbial role, mφs play a crucial role in iron recycling [47]. Mφ lysosomes are the first organelles to receive extracellular iron and are responsible for its reduction into the ferrous (Fe^2+^) form for translocation, transportation for iron recycling or storage. The activity of the iron-processing enzymes in the lysosomes depends on an acidic lysosomal pH [47,48]. Studies have suggested that CFTR plays role in lysosomal pH homeostasis and that CFTR^−^^/−^ leads to alkalinization [34,36,49,50,51,52,53]. This CFTR^−^^/−^-related alkalinization could cause an increase in mφ intracellular iron, which could be further increased by the iron overload observed in the transplant microenvironment [44]. At present, little is known about the functional sequelae that can result from mφs overloaded with iron. We hypothesized that the CFTR^−^^/−^-related mφ impairment in lysosomal acidification would interact with the iron-overloaded lung transplant microenvironment to contribute to the higher rates of invasive *Af* infections in CF-LTRs. To study the transplanted CF host–*Af* relationship, we studied *Af* infection in our OTT model using a CFTR^−^^/−^ mouse as the transplant recipient compared to wild type (WT) control transplant recipients. We examined the impact of pH alkalinization in mφ’s ability to clear *Af* conidia and evaluated the contributory role of iron. By elucidating the role of pH modulation in mφ’s ability to clear *Af* conidia this model has the potential to facilitate the development of novel therapeutic concepts for the treatment of invasive *Af* infections in CF-LTRs.

## 2. Materials and Methods

### 2.1. Aspergillus Culture

Dr. David Stevens provided the Aspergillus fumigatus 10AF strain and Dr. Tobias Hohl provided the *Aspergillus fumigatus* Af293 dsRed fluorescent *Aspergillus* reporter (FLARE) strain. The FLARE strain was developed to trace conidial fate and enable functional analysis of conidiacidal activity [54]. Aspergillus fumigatus conidia were grown in potato dextrose agar plates at 37 °C. Conidia were harvested by washing the surface of the agar plate with 0.05% Tween 80 in saline (v/v). The conidial suspension was vortexed to disperse clumps of conidia and stored at 4 °C overnight (14–16 h). The suspension was then diluted to the desired number of conidia per ml. For in vitro assays, the cell to FLARE conidia ratio used was 1:5 and for in vivo experiments 108 10AF conidia/ml were used.

### 2.2. FLARE Conidia AF633 Tagging

To generate a fluorescent cover to the FLARE conidia, 5 × 10^8^ *Af*293-dsRed conidia were rotated in 0.5 mg/mL Biotin *XX*, SSE (B-6352; Invitrogen) in 1 mL 50 mM carbonate buffer (pH 8.3) for 2 h at 4 °C, washed in 0.1 M Tris-HCl (pH 8), labeled with 0.02 mg/mL AF633-streptavidin (S-21375; Invitrogen) in 1 mL PBS for 30 min at RT, and resuspended in PBS and 0.025% Tween 20 for experimental use [54].

### 2.3. Orthotopic Tracheal Transplantation Model

Five-week-old male C57BL/6, BALB/c and *Cftr^tm1Unc^* Tg(FABPCFTR)1Jaw (CFTR^−/−^) mice were purchased from the Jackson Laboratory. Mice were randomly assigned to groups that consisted of ≥5 transplanted mice in all experiments. The OTT model was used, as previously described [17,44]. Animals were inoculated intratracheally with 4 × 10^6^ *A. fumigatus conidia* on day 7 posttransplant, after which all infected animals received a single dose of corticosteroid (1 mg of triamcinolone acetonide) subcutaneously. At day 3 post infection, all mice were euthanatized using CO_2_ asphyxia and cervical dislocation. The animal protocol used for this study has been approved by Stanford University Research Compliance Office (RCO) Administrative Panel on Laboratory Animal Care (APLAC), active protocol # 32936.

### 2.4. Tissue Preparation, Grading of Fungal Invasion

The extent of fungal invasion was semiquantitatively graded, as previously described [44]. All tracheal samples were cut longitudinally in 5 μm sections through the entire tracheal segment and stained with Grocott’s methenamine silver (Histo-Tec Laboratories). Tracheal sections were graded for depth of fungal invasion. The degree of fungal burden was determined using the following scoring system: 0, no fungal elements; 1, fungal hyphae present in the epithelial layer; 2, hyphae present in the subepithelial layer; 3, hyphae present in the tracheal ring area; and 4, hyphae growth beyond the tracheal ring.

### 2.5. Lung Macrophages Isolation

To isolate lung mφs we used a well-established protocol described elsewhere [55]. Briefly, mice were sacrificed, and lungs were isolated and minced with a scalpel to <1 mm pieces. Minced lung tissue was incubated with Collagenase I (300 μg/mL) and DNase I (5 U/mL) at 37 °C incubator for 25 min. The dissociated lung tissue was passed through a 100 μm cell strainer. The strainer was washed using a tissue wash buffer (PBS, 2% FBS, 2 mM EDTA). To obtain monolayers of mφs, the cell concentration is adjusted to 2–3 × 10^6^ total nucleated cells/mL in DMEM medium. The cells were allowed to adhere to the substrate by culturing them for 1 to 2 h at 37 °C. Non-adherent cells were removed by gently washing three times with warm PBS. At this time, cells should be greater than 90% mφs [56].

### 2.6. Tracheal Tissue Staining

Fresh tracheal cryosections from mice infected with *Af*-GFP were fixed with 4% paraformaldehyde in PBS pH 7.4 for 15 min at room temperature. Tissue was then permeabilized with 0.1% Triton *X*-100 in PBS for 10 min at room temperature. Then, tracheas were then incubated with 1% BSA, 22.52 mg/mL glycine in PBST (PBS + 0.1% Tween 20) for 30 min to block unspecific binding of the antibodies. Tissue was incubated with primary antibody ferritin (ab69090; Abcam, Cambridge, UK) in 1% BSA in PBST for 1 h at room temperature. Following the tissue was incubated with secondary antibody goat anti-Rabbit IgG Texas Red (ab6719; Abcam, Cambridge, UK) in 1% BSA for 1 h at room temperature in the dark. Tracheas were then mounted using ProLong Gold (P10144; Invitrogen^TM,^,Waltham, MA, USA). Images and z stacks were taken using a confocal microscope (Leica Stellaris 8). Fluorescence was quantified by ImageJ.

### 2.7. Conidial Ingestion and Clearance Detection Imaging

Lung mφs were seeded in sterile coverslips. Mφs were allowed to adhere by incubating at 37 °C with 5% CO_2_ overnight. Iron-dextran (D8517; Millipore Sigma, Burlington, MA, USA) in complete media treatments were added for 16 h. FLARE-AF633 conidia were added to each well (cell–conidia ratio 1:5) and samples were incubated for 30 min and 6 h at 37 °C with 5% CO_2_ for the ingestion and clearance assessments, respectively. Cells were then washed 4 times with PBS to remove any unbound conidia and fixed using 4% paraformaldehyde in PBS for 10 min at room temperature. Cells were permeabilized using 0.1% Triton *X*-100 for 10min at room temperature. Finally, samples were then mounted using ProLong Gold with DAPI (P36941; Invitrogen^TM^, Waltham, MA, USA). Images and z stacks were taken using a confocal microscope (Leica Stellaris 8, Leica, Wetzlar, Germany).

### 2.8. Flow Cytometry Methods

Lung mφs were seeded in 6-well plates. Mφs were allowed to adhere by incubating at 37 °C with 5% CO_2_ overnight. Iron-dextran in complete media treatments were added for 16 h. FLARE-AF633 conidia (cell–conidia ratio 1:5) were added to each flask and samples were incubated for 6 h at 37 °C with 5% CO_2_. Cells were then detached using mφs detachment solution DXF (C-41330; PromoCell, Heidelberg, Germany). Cells were washed 3 times with PBS and analyzed immediately by flow cytometry (BD LSR II or FACSymphony, Becton, Dickinson and Company, Franklin Lakes, NJ, USA). After the exclusion of doublets and debris, initial gating was done using the forward-scatter and side-scatter dot plot. Cells were then gated for dsRed+ AF633+ (live conidia) and AF633+ (dead conidia) (Figure A1). Analyses were performed with FlowJo v10.6.2 (FlowJo, Ashland, OR, USA).

### 2.9. Determination of Lysosomal Alkalinization

Lung mφs were seeded in sterile coverslips. Mφs were allowed to adhere by incubating at 37 °C with 5% CO2 overnight. To detect lysosomal acidification in live cells LysoTracker^TM^ Green DND-26 (L7526, Invitrogen, Waltham, MA, USA) 75nM was added to each sample. Cells were then incubated for 30 min at 37 °C with 5% CO2. Cells were then washed and analyzed by flow cytometry (BD LSR II or FACSymphony). Analyses were per-formed with FlowJo v10.6.2 (FlowJo, Ashland, OR, USA).

### 2.10. Cellular pH Measurement

For cellular organelles pH measurements, we used LysoSensor^TM^ Yellow/Blue dextran (L22460; Invitrogen, Waltham, MA, USA) probe. To form a standard curve, murine lung mφs were treated with media with increasing pH from 4 to 8 for 1 h at 37 °C. Then, LysoSensor^TM^ probe 5 mg/mL was added in each sample and cells were incubated at 37 °C for 1 h. Cells were then washed with PBS and images were taken using a confocal microscope (Leica Stellaris 8, Leica, Wetzlar, Germany) at excitation 405 nm and 450 nm and emission 505 and 530 nm, respectively, or fluorescence was detected using a plate reader (Synergy LX Multi-Mode Plate Reader, BioTek, Winooski, VT, USA). pH was calculated based on the standard curve (Figure A2).

### 2.11. Iron Assay

Iron levels were detected using a colorimetric assay (Iron Assay Kit ab83366; Abcam, Cambridge, UK). Tracheas were harvested and washed thoroughly with PBS. Tissue was minced and homogenized using a homogenizer. Tissue homogenate was centrifuged at 16,000× *g* for 10 min, and supernatant was used in this assay according to manufacturer’s instructions (Figure A3).

### 2.12. pH Modulation Treatments

Chloroquine diphosphate (CQ) (41-095-0, Tocris Bioscience™, Bristol, UK) powder was diluted in PBS and used to increase lysosomal pH in primary lung mφs. CFTR^−/−^ and WT mouse lung mφs were treated with increasing concentrations of CQ (20 μM, 40 μM and 80 μM) for 2 h. To decrease lysosomal pH we used Poly(lactic-co-glycolic acid) (PLGA) nanoparticles (NPs) with size ~100 nm (805092, Millipore Sigma, Burlington, MA, USA). PLGA NPs were diluted in PBS. CFTR^−/−^ and WT mouse lung mφs were treated with 0.5 mg/mL NPs for 1 h. After treatments cells were washed three timeswith Dulbecco’s phosphate-buffered saline 1×.

### 2.13. Statistics

GraphPad Prism version 9.0 (GraphPad, San Diego, CA, USA) was used for statistical analysis. Differences in ferritin levels and iron levels were evaluated using Student’s *t* test. All t tests were two-tailed. Histologic differences in the depth of fungal invasion, fungal burden and Prussian blue staining for iron content in the graft were evaluated by a nonparametric Mann–Whitney U test. Results from ingestion and phagocytosis assays were analyzed by one-way analysis of variance (ANOVA) followed by Tukey’s post hoc test for multi-ple comparisons. PLGA studies were evaluated by Student’s t test. Significance values were set at *p* < 0.05.

## 3. Results

### 3.1. In the CF-OTT Model Both Tracheas and CFTR^-/-^ mφs Had Higher Levels of Iron Compared to WT Control

In order to investigate the risk factors that lead to *Af* invasion in CF-LTRs, we developed a CF-OTT model by transplanting a trachea from a Balb/c mouse into a CFTR^−/−^ recipient mouse (*Cftr^tm1Unc^* Tg(FABPCFTR)1Jaw on a C57Bl/6 background). Balb/c mouse tracheas were also transplanted into WT (C57Bl/6) mice and studied as controls. We first evaluated the iron content indirectly by staining the tracheal tissue with an anti-ferritin antibody (Ab) (Figure 1A,B) and directly via an iron detection colorimetric assay (Figure 1C) in non-transplant and transplanted mice. CFTR^−/−^ trachea had a significantly higher tissue iron content than WT trachea (*p* < 0.05). This finding was also seen in explanted trachea from CF-transplant recipients compared to WT controls (*p* < 0.05). To characterize the levels of iron in lung mφs we measured iron levels in CFTR^−/−^ mφs compared to WT mφs. In these studies, we found that iron, particularly in its ferric (Fe^3+^) form was elevated in CFTR^−/−^ lung mφs (Figure 1D). Together, these data suggest that CFTR^−/−^ was associated with high iron levels and that this baseline aberrancy in iron handling persisted in the CF-OTT recipient.

#### 3.1.1. Af Was Significantly More Invasive in CF-OTT Recipients

To study *Af* invasion after airway transplantation, CF-OTT and WT transplants were performed. OTT recipients were infected on day 9 post-transplantation with *Af* conidia, and mice were sacrificed 3 days post-infection (Figure 2A). Explanted trachea were sectioned and studied for *Af* invasion as previously described [17]. In these studies, *Af* consistently demonstrated the highest level of invasion in CF-OTT compared to infections in WT-OTT (*p* < 0.05, Figure 2B,C).

#### 3.1.2. CFTR^−/−^ mφs Have Impaired Ability to Phagocytose and Kill Af Conidia and Are Further Impaired by the Addition of Iron

Given the central role of mφs in iron handling and control of *Af* invasion, we were particularly interested in examining the role of the CFTR^−/−^ iron laden mφs in invasive aspergillosis. To study CFTR^−/−^ mφs’ ability to phagocytose and kill *Af* conidia and the impact of iron, we used the *Af*293 DsRed strain, also known as fluorescent *Aspergillus* reporter (FLARE) strain [54]. The FLARE strain was developed to trace conidial fate and enable functional analysis [54]. The expressed DsRed is used as a viability indicator while the attached fluorescent tracer (Alexa Fluor 633; AF633) allows the detection of conidia after conidial death (Figure 3A). We discovered that in vitro CFTR^-/-^ lung mφs had a decreased ability to phagocytose conidia (CFTR^−/−^: 66.5% vs. WT: 97.5%, *p* < 0.0001, Figure 3B,C). Moreover, the ratio of live to dead, phagocytosed conidia at 6 h was significantly higher in CFTR^−/−^ lung mφs (*p* < 0.05, Figure 3D,E). This dysfunction was impaired in a dose-dependent fashion by the addition of iron (*p* < 0.0001, Figure 3B–E).

#### 3.1.3. CFTR Deficiency, Iron and Treatment with Chloroquin Increase mφ Lysosomal pH

The ability of lysosomes to achieve an acidic pH is essential for iron metabolism and *Af* conidia killing [57,58,59,60]. Although the data are equivocal, studies have suggested that CFTR deficiency may lead to mφ lysosomal alkalinization. To evaluate this potential mechanism, murine WT and CFTR^−/−^ lung mφs were isolated and pH measured using LysoSensor Yellow/Blue dextran. In CFTR^−/−^ mφs, the pH was significantly higher (*p* < 0.05) than the WT mφs (Figure 4A). Moreover, the addition of iron in culture significantly increased the pH of CFTR^−/−^ mφs (Figure A4). To examine the role of lysosomal alkalinization in the CFTR^−/−^ mφs’ inability to clear *Af* conidia, we treated WT and CFTR^−/−^ mφs with the lysosomal alkalinizing agent chloroquine (CQ) [58,59,60]. As expected, treatment with CQ increased both WT and CFTR^−/−^ mφs’ pH (*p* < 0.05, Figure 4B). Poly lactic-co-glycolic acid (PLGA) nanoparticles (NPs) were used to reverse lysosomal alkalinization, as NP are phagocytosed, localize to the lysosome, degrade and release their acidic components [61,62]. Exposure of CFTR^−/−^ mφs to the PLGA NPs reversed the alkalinization caused by CFTR deficiency (Figure 4C). PLGA NP treatments also reversed mφs’ alkalinization caused by CQ treatments (Figure A5).

#### 3.1.4. CQ Treatment Increased Lysosomal pH in Both WT and CFT^−/−^ mφs and Lowering Their Ability to Phagocytose Kill Af Conidia

Having identified reagents that can modulate mφ pH in vitro, we investigated the impact of mφ pH modulation on *Af* conidia phagocytosis and killing. We measured lysosomal acidification after the addition of CQ (20 μM, 40 μM and 80 μM) by Lysotracker^TM^ Green staining followed by flow cytometry analysis (Figure 5A). There was significant difference in lysosomal acidification (*p* < 0.05) between CFTR^−/−^ and WT mφs at baseline (Figure 5A), with no detectable acidic lysosomes in mφs treated with 80 µM of CQ (Figure 5A). We then tested CFTR^−/−^ and WT mφs’ ability to clear *Af* conidia before and after CQ treatments. We show that both CFTR^−/−^ and WT CQ alkalinized mφs have a decreased ability to both phagocytose (Figure 5B) and kill *Af* conidia (Figure 5C). The impact of CQ on CFTR^−/−^ mφs’ ability to kill *Af* conidia was significantly higher (*p* < 0.05) compared to the WT mφs (Figure 5C).

#### 3.1.5. PLGA NPs Restore CFTR^−/−^ mφ Lysosomal Acidity and Ability to Kill Af Conidia

Next, we examined if PLGA NPs treatments can improve CFTR^−/−^ mφ fungicidal function. Treatment of CFTR^−/−^ mφ with PLGA NPs significantly increased (*p* < 0.0001) and restored the percentage of CFTR^−/−^ mφs with acidified lysosomes (Figure 6A). Furthermore, PLGA NPs significantly increased *Af* conidial phagocytosis (*p* < 0.0001) and killing (*p* < 0.05) by the CFTR^−/−^ mφs (Figure 6B,C). Furthermore, PLGA NPs treatments to WT macrophages did not impact their ability to phagocytose conidia, but significantly (*p* < 0.05) increased conidial killing (Figure A6). Combined, these data support our hypothesis that CFTR-related mφs’ alkalinization could be critical factor in the increased risk of invasive aspergillosis in CF-LTRs (Figure 7).

## 4. Discussion

We identified that graft iron overload and CFTR^−/−^-related mφ alkalinization drive *Af* invasion in the CF-OTT model. We showed that (i) iron overload has an impact on CFTR^−/−^ mφ alkalinization and antimicrobial function and (ii) that by correcting lysosomal acidity in CFTR^−/−^ mφs, we can restore their ability to clear *Af* conidia.

*Aspergillus*-related pulmonary disease results in significant morbidity and mortality worldwide [2,3]. Patients with CF are prone to invasive aspergillosis after lung transplantation, with particularly high rates of saprophytic fungal infections of the transplant airway anastomosis [63,64,65]. Other than the risks associated with postoperative impairment of mucociliary clearance after lung transplant and immune suppression little is known of the risk factors that promote these infections in LTRs. Using the OTT model of *Af* infection, we have shown that iron plays a determinant role in the switch from *Af* colonization to invasion in transplantation [44]. Furthermore, we and others have shown that abnormal airway iron homeostasis is a feature of CF lung disease [66,67,68]. It is known that CF patient’s sputum has elevated iron levels [66] and we have identified iron-laden mφs in the biopsies of transplanted airways in CF patients [44].

In order to investigate the impact of the transplant microenvironment on CFTR^−/−^ mφs, we developed a CF-OTT model. In this model the CFTR^−/−^ trachea iron levels were significantly higher (*p* < 0.05) compared to WT at baseline. After the allotransplantion the trachea iron content was also significantly higher (*p* < 0.05) in explanted transplants from CFTR^−/−^ recipients compared to explanted tracheal grafts from the WT transplant recipient. Together, these results suggest that CF-related defects in iron metabolism exist at baseline and that these defects persist in the CF-transplant recipients. These results are consistent with results of other studies suggesting that CFTR-deficiency leads to defects in iron metabolisms and that the iron imbalance after transplantation is more severe in the CFTR^−/−^ airways [44].

Here, we have shown for the first time that CFTR^−/−^ mφs have a decreased ability to kill *Af* conidia and that iron-overload contributes to this impairment. We hypothesize that this inability to clear *Af* is due to CFTR^−/−^-related mφs’ alkalinization. While alkalinization has been described in CFTR^−/−^ mφs and suggested to impair pathogen clearance, other studies have failed to replicate these findings [69,70,71,72]. Mφs act as the first line of defense by recognizing and engulfing pathogens in phagosomes that subsequently mature into phagolysosomes, where the microorganisms are cleared [73,74]. It has been reported that alveolar mφs from CFTR-null mice were unable to kill bacteria [71]. pH in phagolysosomes in alveolar mφs from CFTR-null mice was ∼2 pH units more alkaline than those from control mice and there was a defect in lysosomal acidification in CFTR-null alveolar mφs, with pH ∼ 6 in CF mφs vs. ∼4.5 in control mφs, which was proposed, by a delayed phagolysosomal fusion mechanism, accounting for the defect in acidification [71]. Although, there is contradicting evidence regarding the connection between CFTR deficiency and mφ lysosomal acidification, there is clear evidence that dysregulation of lysosomal acidification in mφs impairs their antimicrobial functions [70]. Our findings demonstrated that CFTR^−/−^ mφs are more alkaline (pH ~ 7.9) than WT control mφs (pH ~ 7, Figure 4A) and that iron addition significantly increased this alkalinization. These data suggest that increased iron overload could contribute to the alkalinization of the CFTR^−/−^ mφs.

To further investigate the role of pH modulation in mφs anti-microbial function, we treated WT and CFTR^−/−^ macrophages with CQ a known lysosomotropic agent that increases lysosomal pH by accumulating within lysosomes as a deprotonated weak base [75]. CFTR^−/−^ and WT mφs treated with CQ showed an inability to kill *Af* conidia in a dose-dependent fashion and correlated with the loss of lysosomal acidification. These data suggest that the increase in lysosomal pH could be a factor that promotes *Af* invasion. To confirm these findings, we used PLGA NPs to acidify the lysosomes of the CFTR^−/−^ and the WT mφs. PLGA is a biocompatible and biodegradable polymer that degrades in the acidic environment of the lysosome and releases lactic and glycolic component carboxylic acids with pKas of 3.86 and 3.83, respectively, resulting in lysosomal acidification [61]. In our studies, PLGA NPs treatments restored CFTR^−/−^ mφs acidity and the ability to kill *Af* conidia. These data strongly suggest that CFTR^−/−^ mφ pH is a key modulator of their anti-microbial function.

One limitation of this study was an inability to quantify mφ lysosomal pH in vivo resulting from technical difficulties. In preliminary studies using flow cytometry, we were not able to delineate between auto fluorescence of the transplant tissue and the LysoSensor probes. However, we were able to isolate mφs from the CFTR^−/−^ mouse lungs and measure their pH in vitro. Additionally, controversy currently exists regarding the exact mode of *Af* invasion with in vitro studies demonstrating a variety of potential pathways including: (i) damage of epithelial cells and direct hyphal invasion; (ii) induction of epithelial cell damage by fungal proteases the result in disruption of tight junctions; and (iii) internalization of fungal conidia and subsequent escape [76,77]. Such a limitation may be overcome by studying serial time points after inoculation of *Af* conidia using transmission electron microscopy [77]. However, as TEM is notoriously insensitive and would require a high number of transplants, such studies were considered beyond the scope of the current work. Furthermore, the effect of PLGA NPs treatment in *Af* invasion was not examined in vivo since a detailed investigation of their route of administration, absorption, distribution, metabolism and excretion (ADME) and toxicity in mice would be necessary to adequately dose this medication. Future studies will examine the possibility of PLGA NP treatment to decrease *Af* invasion in the CF-OTT model. In addition, we recognize the central role of neutrophils in *Af* clearance. As CFTR is known to be present on multiple immune cell subtypes, dendritic cells, monocytes/macrophages, neutrophils and lymphocytes [28,37,76,77], future studies should also examine the role of CFTR deficiency on neutrophil function in *Af* clearance.

For the first time, we show that iron-laden CFTR^−/−^ mφs are unable to kill *Af* conidia as a result of CFTR^−/−^-related lysosomal alkalinization and suggest that the CFTR^−/−^ mφs may be a critical effector cell in the *Af* graft invasion. Although limited to primary cell culture, we have shown in a dose-dependent manner that both phagocytosis and the ratio of *Af* conidia that remained alive in CFTR^−/−^ mφs increased with iron or CQ treatment addition. Together, our data suggest that in the lung transplant microenvironment, the CFTR^−/−^-related mφs’ lysosomal alkalinization is increased and leads to an inability to clear *Af* conidia. These findings may be clinically significant as mφ pH modulation could represent an actionable therapeutic target against *Af* invasion.

## Figures and Tables

**Figure 1 jof-08-00751-f001:**
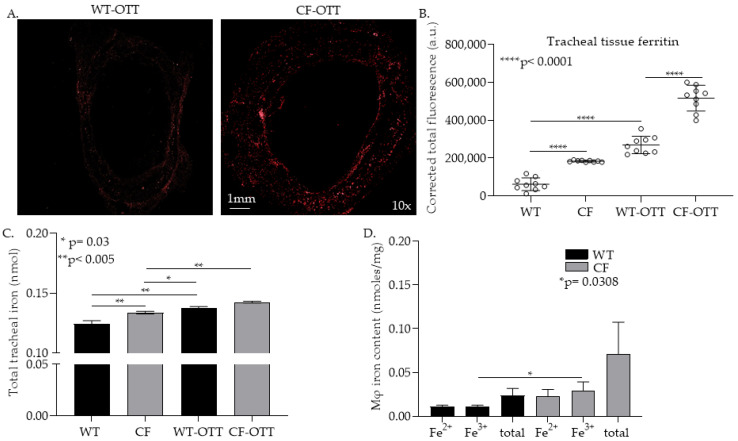
In the CF OTT model, both CF tracheas and macrophages had higher levels of iron compared to WT control. (**A**) Representative image of explanted tracheal transplant tissue from wild type (WT) transplant recipient (**left**) and CF-transplant recipient (**right**), stained for ferritin. (**B**) Quantification of detected fluorescence in trachea from non-transplanted WT and CF mice and allotransplants in WT and CF recipients (**** *p* < 0.0001) (*n* = 5/group). (**C**) Quantification of total tracheal iron in non-transplanted WT and CF mice and allotransplants in WT and CF recipients (* *p* = 0.03, ** *p* < 0.005) (*n* = 5/group). (**D**) Quantification iron in mφs isolated from mouse tracheas (*n* = 3/group) (* *p* = 0.0308).

**Figure 2 jof-08-00751-f002:**
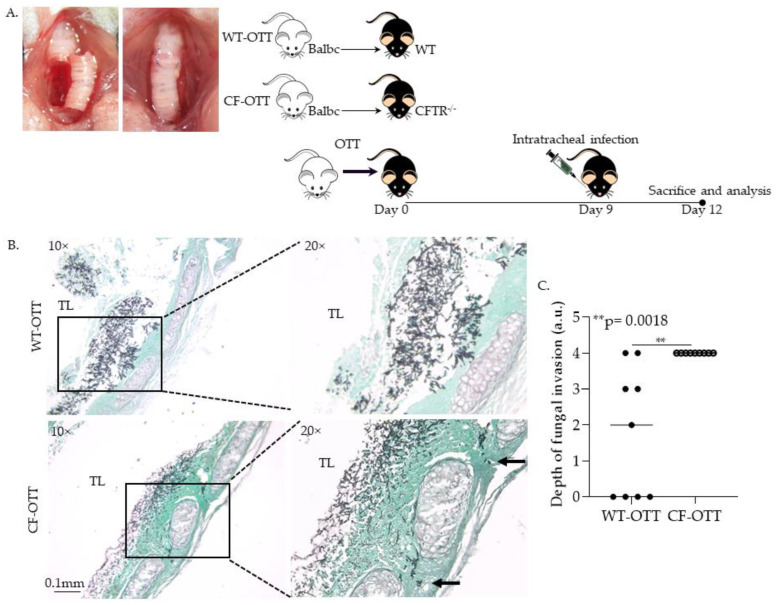
*Af* was significantly more invasive in the CF orthotopic tracheal transplant (OTT) compared to the WT-OTT model. (**A**) Representative picture of OTT surgery (**right**) and schematic picture (**left**) of WT-OTT and CF-OTT model of *Af* infection. (**B**) Explanted tracheal tissue sections from WT-OTT (**top**) and CF-OTT (**bottom**) stained with Grocott Methenamine Silver stain for fungal elements. (**C**) Quantification of *Af* invasion in the WT-OTT and CF OTT models (*n* = 10/group) ** *p* = 0.0018) (TL: tracheal lumen).

**Figure 3 jof-08-00751-f003:**
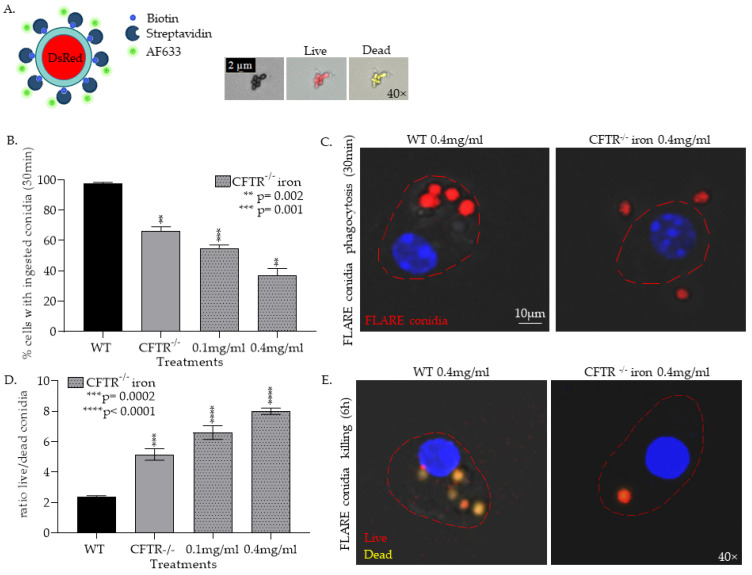
CFTR^−/−^ mφs have an impaired ability to phagocytose and kill *Af* conidia. (**A**) Illustration of DsRed expressing conidia labeled with AF633 and confocal images of labeled conidia. (**B**) Quantification of WT and CFTR lung mφ ability to phagocytose conidia as measured by flow cytometry (** *p* = 0.002, *** *p* = 0.001). (**C**) Representative confocal microscopy image of mφ phagocytosis of FLARE conidia (**D**) Quantification of WT and CFTR lung mφ ability to kill conidia as measured by flow cytometry (*** *p* = 0.0002, **** *p* < 0.0001). (**E**) Representative confocal microscopy image of FLARE conidia killing.

**Figure 4 jof-08-00751-f004:**
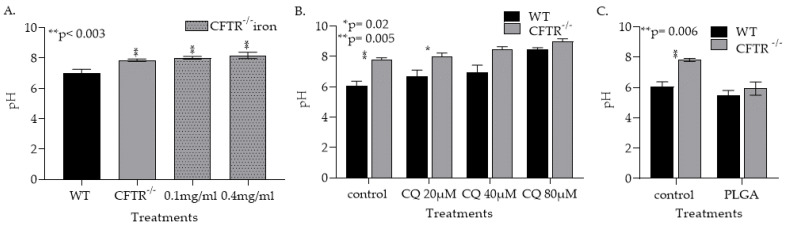
CFTR deficiency, iron and treatment with chloroquine (CQ) increase mφ lysosomal pH. (**A**) Measurement of pH in WT and CFTR lung mφ by LysoSensor^TM^ Yellow/Blue dextran staining (** *p* < 0.003). (**B**) Measurement of pH in WT and CFTR lung mφ treated with CQ by LysoSensor^TM^ Yellow/Blue dextran staining (* *p* = 0.02, ** *p* = 0.005). (**C**) Measurement of pH in WT and CFTR lung mφ by LysoSensor^TM^ Yellow/Blue dextran staining in CFTR^−/−^ and WT mφs after poly lactic-co-glycolic acid (PLGA) treatment (** *p* = 0.006).

**Figure 5 jof-08-00751-f005:**
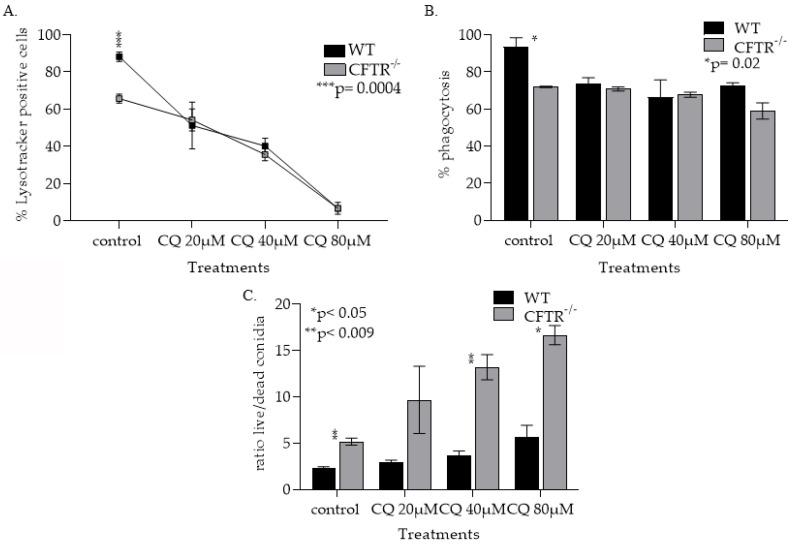
CQ treatments increased lysosomal mφ pH and decreased mφ ability to phagocytose and kill *Af* conidia. (**A**) Measurement of lysosomal acidity in WT and CFTR lung mφ treated with CQ, using Lysotracker^TM^ (*** *p* = 0.0004). (**B**) Quantification of the ability of WT and CFTR lung mφs treated with CQ to phagocytose *Af* conidia as measured by flow cytometry (* *p* = 0.02). (**C**) Quantification of the ability of WT and CFTR lung mφs treated with CQ to kill *Af* conidia, as measured by flow cytometry (* *p* < 0.05, ** *p* < 0.009).

**Figure 6 jof-08-00751-f006:**
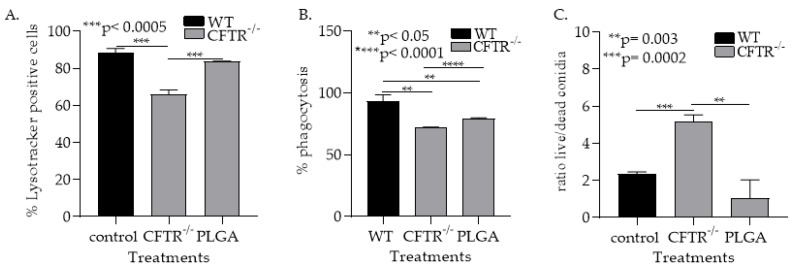
PLGA NPs restore lysosomal acidity and mφ ability to clear *Af* conidia. (**A**) Measurement of lysosomal acidity in WT and CFTR lung mφ treated with PLGA, using Lysotracker^TM^ (*** *p* < 0.0005). (**B**) Quantification of the ability of WT and CFTR lung mφs treated with PLGA to phagocytose *Af* conidia as measured by flow cytometry (** *p* < 0.05, **** *p* < 0.0001). (**C**) Quantification of the ability of WT and CFTR lung mφs treated with PLGA to kill *Af* conidia, as measured by flow cytometry (** *p* = 0.003, *** *p* = 0.0002).

**Figure 7 jof-08-00751-f007:**
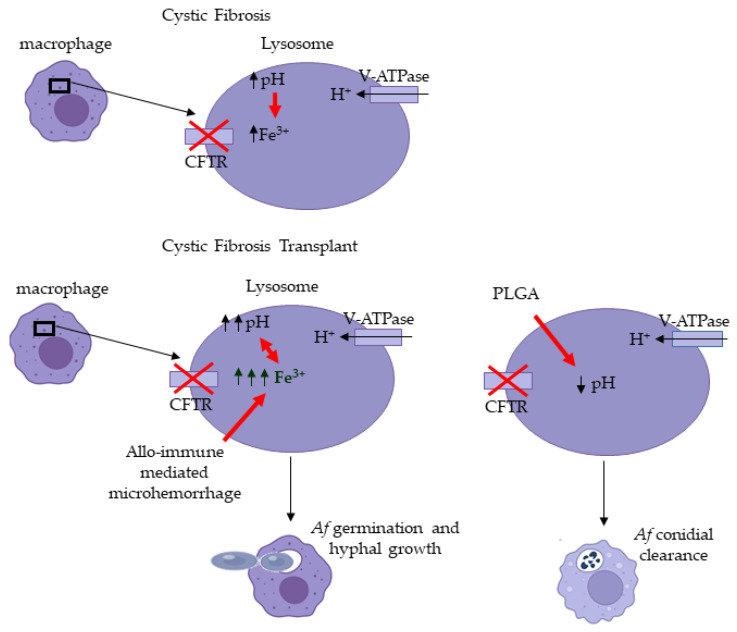
Graphical Conclusions. Loss of CFTR increased mφs’ lysosomal pH. Iron overload seems to have an additive effect to the CFTR^−/−^ mφs’ alkalinization. PLGA NPs can restore the pH acidity and increase *Af* conidial killing.

## Data Availability

Not applicable.

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
