# Peer review of "Macrophage Lysosomal Alkalinization Drives Invasive Aspergillosis in a Mouse Cystic Fibrosis Model of Airway Transplantation"

_jof, 2022, doi:10.3390/jof8070751_

Round 1
Reviewer 1 Report
The article as presented was a well written and interesting series of experiments with some valid conclusions about the effect of lysosomal alkalinsation of murine macrophage phagocytosis and killing of Aspergillus conidia.
However the link between these findings and iron levels in the lung and relevancy to the transplant field specifically are not clear. Additionally as all the work is in murine primary cells the effects in human disease are not proven.
To improve the paper, I would ideallyl like to see experimental evidence of:
Effect of iron supplementation on wild type lung macrophage lysosomal pH.
Effect of iron supplementation on wild type lung macrophage phagocytosis and killing of conidia.
Effect of adding PGLA to WT macrophages, in terms of phagocytosis and conidial killing, to show no 'off target' effects.
Could the authors speculate as to why chloroquine has an effect on reducing phagocytosis in wild type macrophages given its primary effect in these experiments is to reduce lysosomal acidification. This suggests that chloroquine may have other effects in this model that may confound the related conclusions.
The effect of immunosuppression in the setting of a transplant is not addressed. This may differentially affect the function of macrophages when comparing WT and CFTR-/- cells.
Ideally some of this work should be replicated in human macrophages, even if cell line, and preferentially primary human cells.
Lines 369-71 - I presume this sentence should be removed?
Reviewer 2 Report
The authors standardised a murine CF tracheal transplant model to determine the effect of macrophage lysosomal alkalinisation in their ability to eliminate Aspergillus fumigatus conidia. Using in vivo and in vitro approaches, they demonstrated that internal pH is critical to iron recycling, as well as per lysosomal activation and killing of Aspergillus conidia . They also showed that phagocytosis could be enhanced or impaired after the modification of the pH microenvironment, in both wildtype and CFTR-/- macrophages.
Overall, this manuscript is worth to be published, the scientific approach and the methodology are sound and the results and discussion seem robust. Nevertheless, some improvements must be done, specially in the material and methods section and minor typos and grammar errors must be corrected:
Page 2 line 94. Further information regarding Af strains should be provided, p.ex. what does FLARE stands for ?, information provided in the results section should be included here.
Page 3 line 97. information regarding time length of conidia conservation at 4° C is needed since conidia swelling can happen.
Page 3 line 98. information regarding which Af strain was used to in vitro or in vivo assays should be included.
Page 3 line 113. Infection dose is lacking.
Page 4 line 184. No protocol regarding the use of Lysotracker has been found. It has to be included (at the pH modulation treatment part).
Page 5 line 207. CQ diluent is lacking. The same lacking information was observed for PLGA.
Page 5 line 248. Delete double parenthesis
Information regarding statistical tools used must been included
Figure 4. Decrease of phagocytic activity at increasing CQ concentrations does not seem as straight as decrease of Lysotracker +cells, could you further discuss?
Figure 6B PLGA treatment seems not able to increase phagocytosis to "normal "levels" while lysotracker does not indicate significant differences between control and PLGA macrophages, could you further discuss?
Page 11 line 368. There is a missing reference and 3 text lines are out of context.
Reference 44 must be corrected.
Reviewer 3 Report
The authors investigated the role of lysosomal alkalinization as risk factors for invasive aspergillosis. As stated by the authors (line 63-67), it was shown before that the signaling and lysosomal pH in macrophages of CF patients is altered, although there are also contradictory studies. In addition, it was also published before that transplantation results in iron overload, especially in CF patients. Therefore, they investigated the influence of iron on lysosomal alkalinization and the influence of lysosomal pH modification on Aspergillus fumigatus clearance. Nevertheless, there are some major issues regarding the manuscript, which I would like to point out in the following.
Major:
1.) Which iron form does the iron dextran consist of? If I got it correctly, this was used to load the lysosomes with iron. Depending on the iron form (Fe(II), Fe(III), as chlorid, hydroxide, etc.) the metal itself can influence the pH, either acidify or alkalinize. Therefore, the conclusion that iron overload leads to alkalinization of lysosomes cannot be drawn without stating the exact iron form.
2.) What is the proposed mode of action, how iron overload leads to alkalinization? An acidic pH is needed to reduce iron to Fe(II) in lysosomes (line 73-75) and therefore, an alkalinized lysosome would lead to an iron increase in macrophages. But why should additional iron increase the pH even more?
3.) Was the influence of iron overload also tested in wt macrophages? Would be interesting to show that iron supplementation is not only displaying a more severe phenotype in the CF macrophages, but also in wt macrophages (Figure 3 and 4).
4.) What is your hypothesis, why increasing the pH in wt lysosomes with chloroquine does reduce the phagocytosis to CF macrophage levels, but the killing is still much more efficient. I would have expect it the other way around that acidic lysosomes are very important for killing. And why is the pH influencing the phagocytosis rate?
Minor:
Line 116 “surviving mice”: Were there also mice, which already died within the first 3 days post infection? An were there differences between wt and CF mice?
Materials and Methods Flow Cytometry method: It would be good to the gating. Especially for phagocytosis, FC is not the optimal method as it will not diminish between adhered and phagocytosed conidia. Displaying the gating would convince the reader more.
In addition, 6h co-incubation is quite long, as already some conidia might have started germinating. Did you see germlings in your cultures and if so, how were these taken into account by FC?
Results 3.1.1: In the M&M section you state that ferritin antibody was used for GFP infected trachea. But in this paragraph, infection is not mentioned. Therefore, it’s not clear to me, if you used here infected or non-infected trachea to determine the iron concentration.
Line 237: It must be Figure 2B and C, not just 2B
Line 272: In the M&M section, it’s not described how you used the Lysotracker. Only the Lysosensor is described there.
Line 287: Phagocytosis is shown in Figure 6B instead of 6C. And you didn’t implement Figure 6C in your text.
Figure1: In Figure 1C, I guess the bar with one digit should show the significance between WT-OTT and CF-OTT and not between CF and WT-OTT, right?
And overall in Figure 1: does the (n=3/group) belong to the whole figure or just for 1D? Please indicate.
Please have a look on all your statistical significance digits in the figures and corresponding text that they are consistent. For example, p < 0.5 is in Figure 1 “*”, in Figure 5A “***” and in Figure 5B “**”.
Line 368 “[ref, STM]”: I guess, you forgot to add a reference here.
Line 369-371: Should this be deleted???
Line 398-400: The degradation of PLGA is highly pH dependent. Therefore I would guess that the degradation in alkalinized lysosomes is much less compared to normal lysosomes. Did you ever check the degradation at the CF lysosomal pH?
Round 2
Reviewer 3 Report
Thanks to the authors for their effort and addressing all my comments. It's a very nice study
Author Response
Dears JoF Editorial Staff and Ms. Zhang,
We appreciate the opportunity to revise our manuscript. We are grateful for the attention given to this study and believe that the revised manuscript is considerably improved.
Yours sincerely,
Efthymia Iliana Matthaiou